# Exploring the design space of visual context representation in Video MLLMs

**Yifan Du**[1]*, **Yuqi Huo**[2]*, **Kun Zhou**[3]*, **Zijia Zhao**[4], **Haoyu Lu**[1], **Han Huang**[4]
**Wayne Xin Zhao**[1]†, **Bingning Wang**[2]†, **Weipeng Chen**[2], **Ji-Rong Wen**[1]
[1]Gaoling School of Artificial Intelligence, Renmin University of China
[2]Baichuan Inc.
[3]School of Information, Renmin University of China
[4]Institute of Automation, Chinese Academy of Sciences
{yifandu1999, batmanfly}@gmail.com
{daniel}@baichuan-inc.com

## Abstract

Video Multimodal Large Language Models (MLLMs) have shown remarkable capability of understanding the video semantics on various downstream tasks. Despite the advancements, there is still a lack of systematic research on visual context representation, which refers to the scheme to select frames from a video and further select the tokens from a frame. In this paper, we explore the design space for visual context representation, and aim to improve the performance of video MLLMs by finding more effective representation schemes. Firstly, we formulate the task of visual context representation as a constrained optimization problem, and model the language modeling loss as a function of the number of frames and the number of embeddings (or tokens) per frame, given the maximum visual context window size. Then, we explore the scaling effects in frame selection and token selection respectively, and fit the corresponding function curve by conducting extensive empirical experiments. We examine the effectiveness of typical selection strategies and present empirical findings to determine the two factors. Furthermore, we study the joint effect of frame selection and token selection, and derive the optimal formula for determining the two factors. We demonstrate that the derived optimal settings show alignment with the best-performed results of empirical experiments. The data and code are available at: https://github.com/RUCAIBox/Opt-Visor.

## 1 Introduction

Recent advancements in video Multimodal Large Language Models (*video MLLMs*) have shown the great potential in extending LLMs (Zhao et al., 2023) to process video data (Lin et al., 2023). Typically, a video MLLM is developed based on a pre-trained LLM, and an image encoder will be attached to the LLM via a modality projector, which links the textual and visual semantic spaces. In this way, we can prompt the video MLLM with textual instruction and visual embeddings, to generate the natural language response for fulfilling the video-based task, *e.g.,* video question answering (Xu et al., 2017) and video captioning (Caba Heilbron et al., 2015). Despite the success, it is still challenging for existing video MLLMs to handle complex or long videos, due to the limited model capacities.

To develop effective video MLLMs, previous research work mainly focuses on two aspects, either improving the model architecture (Wang et al., 2024b) or enhancing the model training (Zhang et al., 2024b; Liu et al., 2024b). However, another important aspect has been missing in the related literature, *i.e.,* visual context representation. In this work, *visual context* refers to the visual embeddings in the prompt of video MLLMs. Unlike text and images, it is not very straightforward to represent a video. In existing approaches, a widely used way is to sample a number of frames from a video

---

*Equal contribution.
†Corresponding author.

(*frame selection*) and then further sample or generate a number of embeddings for each selected frame (*embedding selection*). However, it is unclear how each factor affects the performance of video MLLMs, and how both factors jointly contribute to the performance improvement within the limited context length of the underlying LLM.

Considering this issue, in this paper, we take the initiative to explore the design space for visual context representation, and derive more effective representation schemes to improve the performance of video MLLMs. Specifically, we firstly formulate the studied task as a constrained optimization problem: given the maximum visual context window size, we model the language modeling loss as a function of the number of frames and the number of embeddings (or tokens) per frame. Such a formulation is useful to help understand the competitive relationships between frame selection and embedding selection. Subsequently, we conduct extensive empirical experiments to explore the scaling effects in frame and embedding selection respectively, and fit the corresponding function to describe the performance trend. Our findings show that: (1) overall increasing the number of visual embeddings (either tokens or frames) would enhance the performance, while scaling the frames can lead to consistently improved performance; (2) the compression-based method can effectively preserve more semantic information with fewer visual embeddings. Furthermore, we study the joint effect of the two factors and propose the method to find the optimal allocation given the limited context length, which is further supported by empirical experiments.

The major contributions of our work are as follows:

- To the best of our knowledge, this is the first work to systematically study the design of visual context, which is an important yet under-explored problem for developing capable video MLLMs. We provide both theoretical formulations and empirical findings to approach this problem.

- We study the scaling effects of model performance *w.r.t.* the number of selected frames and the number of selected embeddings per frame respectively. We fit the corresponding function curve, and compare different strategies (*i.e.,* sampling- and compression-based methods) for both factors.

- We explore the trade-off relationships for frame and embedding selection, and suggest the optimal formula for determining the two factors. We demonstrate that the derived optimal settings show alignment with the best-performed results of empirical experiments.

## 2  PRELIMINARY

In this section, we introduce the background for building the base model in our work.

**Model Architecture.**  Following existing works (Liu et al., 2024a; Zhang et al., 2024c), we adopt the LLaVA-like model architecture, consists of a visual encoder, an LLM, and a projector that maps the visual embeddings to the semantic space of the LLM. Formally, given a video with $T$ frames $\{I_t\}_{t=1}^T$, each frame is encoded by the image encoder $f_\phi$ to obtain $M$ visual embeddings $\{\mathbf{v}_i^t\}_{i=1}^M$, where $\mathbf{v}_i^t \in \mathbb{R}^{d_v}$ denotes the $i$-th visual embedding in the $t$-th frame, and $d_v$ is the dimensionality of the visual embedding. Then the projector $f_\psi$ projects these visual embeddings into the semantic space of the LLM, producing $\mathbf{h}_i^t \in \mathbb{R}^d$.    These visual embeddings are concatenated with the embeddings of a textual prompt $\{\mathbf{e}_j\}_{j=1}^N$, where $\mathbf{e}_j \in \mathbb{R}^d$ is the embedding of the $j$-th token in the prompt. The concatenated sequence is fed as input to the LLM $f_\theta$ to generate the output:

$$y_1 \cdots y_K = f_\theta([\mathbf{h}_1^1, ..., \mathbf{h}_M^1, ..., \mathbf{h}_1^T, ..., \mathbf{h}_M^T, \mathbf{e}_1, ..., \mathbf{e}_N]) \tag{1}$$

During training, we optimize the parameters $\{\phi, \psi, \theta\}$ by minimizing the next-token prediction loss.

**Training Data.**  Based on existing instruction datasets, we mix several widely used image instruction and video instruction sets to construct a new instruction dataset. For the image instruction set, we adopt Cauldron (Laurençon et al., 2024b), which is a large image instruction set based on 50 vision-language datasets. For the video instruction set, we collect the instructions from VideoChatGPT-100K (Muhammad Maaz & Khan, 2023), ShareGPT4Video (Chen et al., 2024), ShareGPTVideo (Zhang et al., 2024b), VIM (Du et al., 2024), as well as some instruction data from VideoChat2 (Li et al., 2024b). The statistics of each instruction set are listed in Table 6.

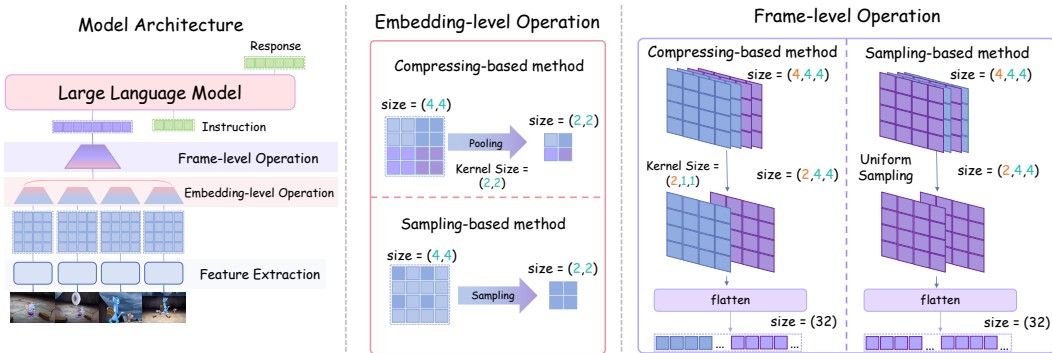

Figure 1: Overview of the LLaVA-like architecture for video-MLLM, and our used frame-level and embedding-level operations for adjusting the visual context window size.

**Implementation Details.** We adopt SigLIP (Zhai et al., 2023) as the image encoder, Qwen2-7B (Yang et al., 2024) as the base LLM, and a two-layer MLP as the projector. We train all the models with the training data listed in Table 6 for 1 epoch. We have tried to include a pre-training stage before the visual instruction tuning, using the 558K pre-training data and only updated the parameters in the MLP following LLaVA (Liu et al., 2024a), but found no obvious difference. All the experiments are conducted on 32 Nvidia H800, with the detailed hyperparameters listed in Table 7.

**Evaluation Setup.** To quantitatively assess the scaling effect of visual context in video MLLMs, we consider the following two metrics for evaluation:

• *Language modeling loss.* It is a continuous measure of model performance in predicting the next token, used to estimate the parameters of the scaling law function. Following Chinchilla (Hoffmann et al., 2022), each model is trained for one epoch to ensure that training samples are unseen when calculating the loss for evaluation. We conduct experiments with five random seeds under a widely used configuration to guarantee the robustness of our results, with detailed results in Appendix E.1.

• *Zero-shot Accuracy.* The zero-shot accuracy can reflect the performance of the model in the real-world application. We select several long video understanding benchmarks for evaluation, including Event-Bench (Du et al., 2024) (only with the challenging episodic reasoning task), VNBench (Zhao et al., 2024), MLVU (Zhou et al., 2024), and VideoMME (Fu et al., 2024). All the questions in these benchmarks are multiple-choice, and we use accuracy as the evaluation metric.

## 3 SCALING LAW OF VISUAL CONTEXT

### 3.1 PROBLEM FORMULATION

As introduced in Section 2, existing video LLMs typically follow the vision-language model architecture (Liu et al., 2024a; Zhang et al., 2024c), which represents a video into multiple representative frames. Further, each frame will be encoded into a number of visual tokens or embeddings. The aggregation of the visual embeddings from all selected frames is referred to as *visual context* in this work. To set the visual context, it is essential to determine two aspects when the base architecture is fixed: (1) how to select the frames from a video (*frame selection*), and (2) how to select the visual embeddings from an input frame (*embedding selection*). Since the base architecture is developed on an existing LLM, the length of visual context is naturally limited by its context length, *i.e.,* the maximum length of input tokens. The two aspects would be competitive in input length allocation: the more the selected frames, the fewer the visual embeddings per selected frame, and vice versa.

In this work, we study the optimal allocation relationship of the visual context for a given video. Formally, we model the language modeling loss $\mathcal{L}(T, M)$ as a function of the number of frames $T$ and the number of embeddings (or tokens) per frame $M$. Given the maximum visual context window size $L$, the number of frames $T$ and visual embeddings per frame $M$ should satisfy the constraint:

$T \times M < L$, we aim to find the optimal solution in minimizing $\mathcal{L}(T, M)$ under this constraint:

$$T_{\text{opt}}(L), M_{\text{opt}}(L) = \underset{T, M \text{ s.t. } T \times M < L}{\operatorname{argmin}} \mathcal{L}(T, M), \tag{2}$$

where $T_{\text{opt}}(L)$ and $M_{\text{opt}}(L)$ represent the optimal allocation strategy for the frame and visual embedding, respectively, with the input limit $L$. To approach it, in the following, we will explore the scaling effect of frame and embedding selection in Section 3.2 and Section 3.3 respectively.

## 3.2 SCALING EFFECT OF THE VISUAL EMBEDDINGS

We first analyze the scaling effect of visual embeddings in a frame for a fixed number of frames. Specifically, we utilize two methods to select (or generate) the visual embeddings in a frame: the sampling- and compression-based method.

### 3.2.1 SAMPLING-BASED METHOD

**Experimental Setup.** In this part, each image is first converted to $27 \times 27$ embeddings by the image encoder, then we vary the number of sampled visual embeddings. Specifically, we uniformly sample $\{1^2, 2^2, 3^2, 4^2, 5^2, 6^2, 7^2, 9^2, 14^2\}$ embeddings from the $27 \times 27$ embeddings, as illustrated in Figure 1. Other sampling methods like block sampling (Li et al., 2023c) will be explored in future work. We set $T = 32$ as the constant, and uniformly sample frames from each video and keep all other factors the same to train 9 video MLLMs with this setup of visual embeddings.

**Fitting Function.** We propose the following function to fit the scaling law of visual embeddings:

$$\mathcal{L}(M) = L_M + \left(\frac{M_0}{M}\right)^{\alpha_M} \tag{3}$$

We fit the language modeling loss with respect to the number of visual embeddings $M$ using the scipy curvefit function, obtaining $L_M = 0.48, M_0 = 1.16 \times 10^{-5}, \alpha_M = 0.1$, with $R^2 = 0.927$ indicating a good fit. The fit curve in Figure 2a shows that $\mathcal{L}(M)$ decreases with increasing $M$, following a power-law like trend. We calculate the mean squared error between the actual loss and predicted loss, obtaining a value of 0.0001, which indicates a very low fitting error.

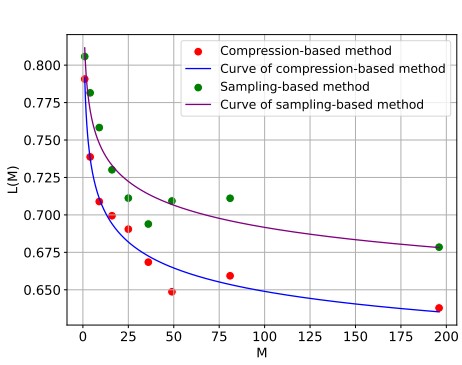

(a) The scaling curve of visual embeddings.

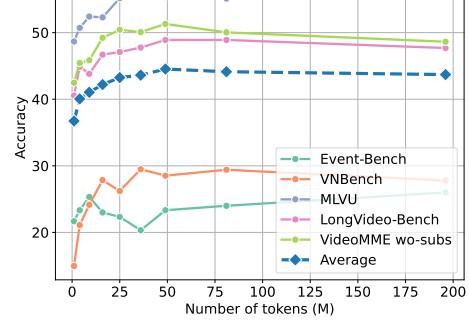

(b) The relationship between the number of embeddings per frame and the benchmark accuracy.

Figure 2: The scaling law of visual embeddings, reflected by the language modeling loss and the zero-shot accuracy on video understanding benchmarks.

**Benchmark Performance Analysis.** Table 1 shows the results of scaling visual embeddings on the evaluation benchmarks. Overall, the model performance improves as the number of visual embeddings increases, especially when it varies from 1 to 4. The improvement becomes more marginal with increasingly more visual embeddings. However, when it exceeds some threshold, the performance starts to decrease. For example, using 196 tokens is worse than using 49 tokens. An

Table 1: Results of sampling-based method under different number of visual tokens per frame.

| # Frames | # Embed./ Frames | Event-Bench | VNBench | MLVU | LongVideo-Bench | VideoMME wo/w-subs | Avg. |
|---|---|---|---|---|---|---|---|
| 32 | 1 | 21.67 | 14.96 | 48.67 | 40.56 | 42.48/52.04 | 36.73 |
| 32 | 4 | 23.33 | 21.11 | 50.72 | 44.88 | 45.44/54.81 | 40.05 |
| 32 | 9 | 25.33 | 24.15 | 52.42 | 43.82 | 45.85/54.67 | 41.04 |
| 32 | 16 | 23.00 | 27.85 | 52.29 | 46.70 | 49.22/57.11 | 42.20 |
| 32 | 25 | 22.33 | 26.22 | 55.18 | 47.08 | 50.44/58.26 | 43.25 |
| 32 | 36 | 20.33 | **29.48** | 56.05 | 47.76 | 50.07/58.04 | 43.62 |
| 32 | 49 | 23.33 | 28.52 | 55.41 | **48.90** | **51.30/59.74** | **44.53** |
| 32 | 81 | 24.00 | 29.41 | 55.12 | **48.90** | 50.04/57.30 | 44.13 |
| 32 | 196 | **26.00** | 27.78 | **56.53** | 47.69 | 48.63/55.59 | 43.70 |

Table 2: Results of compression-based method under different number of visual tokens per frame.

| # Frames | # Embed./ Frames | Event-Bench | VNBench | MLVU | LongVideo-Bench | VideoMME wo/w-subs | Avg. |
|---|---|---|---|---|---|---|---|
| 32 | 1 | 18.67 | 18.44 | 49.45 | 39.88 | 41.33/49.15 | 36.15 |
| 32 | 4 | 24.33 | 27.04 | 52.77 | 43.44 | 46.11/55.93 | 41.60 |
| 32 | 9 | 22.33 | 27.41 | 53.83 | 45.11 | 47.78/55.41 | 41.98 |
| 32 | 16 | 20.33 | 28.96 | 55.04 | 46.32 | 49.85/58.15 | 43.11 |
| 32 | 25 | 23.00 | 28.67 | 54.21 | 46.02 | 49.85/58.11 | 43.31 |
| 32 | 36 | 27.33 | 30.00 | 53.73 | 48.45 | 50.33/58.74 | 44.76 |
| 32 | 49 | 21.33 | 29.93 | 54.84 | 47.16 | 49.96/58.37 | 43.60 |
| 32 | 81 | 23.33 | 27.33 | **57.59** | 48.60 | 52.00/59.37 | 44.70 |
| 32 | 196 | **29.00** | **31.56** | 56.81 | **52.24** | **53.56/59.48** | **47.11** |

interesting finding is that the language modeling loss with 196 embeddings is significantly smaller than that of the model trained with 49 embeddings, as shown in Figure 2a, which indicates that model loss might not directly reflect the performance on downstream tasks.

### 3.2.2 COMPRESSION-BASED METHOD

**Experimental Setup.** We utilize the MeanPooling (Yao et al., 2024a) strategy for compressing the visual embeddings, which has been widely used in visual information processing. Another advantage is that it does not introduce extra parameters, avoiding the influence of new factors in the experiments. We apply MeanPooling with different kernel sizes on the feature map produced by the image encoder and obtain the condensed representation of the image. Specifically, each image is encoded into $27 \times 27$ visual embeddings, on which we apply $p \times p$ mean pooling with stride $p$ ($p = \{2, 3, 4, 5, 6, 7, 9, 14, 27\}$), obtaining $\{1^2, 2^2, 3^2, 4^2, 5^2, 6^2, 7^2, 9^2, 14^2\}$ condensed embeddings per image. All the other factors are kept the same for fair comparison.

**Fitting Function.** We use Equation 3 to fit the scaling law and obtain $L_M = 0.57, M_0 = 0.01, \alpha_M = 0.39$, with $R^2 = 0.987$. The mean squared error between the predicted loss and the actual loss is $5.32 \times 10^{-5}$, indicating a good fit. Compared to the parameters of sampling-based method in Section 3.2.1, where $\alpha_M = 0.1$, the $\alpha_M$ of the compression-based method is significantly larger, implying that increasing the number of embeddings using the compression-based method will result in faster loss decrease, as shown in Figure 2a. Additionally, the compression-based method consistently yields a lower loss than the sampling-based method for the same number of visual embeddings. This is because the compression-based method does not directly discard visual embeddings but instead aggregates information from them, which preserves more spatial information.

**Benchmark Performance Analysis.** For the benchmark evaluation in Table 2, the overall accuracy consistently increases as the number of visual embeddings increases. This finding is significantly different from that in Table 1. This result highlights the advantage of the compression-based method, which can preserve more information than the sampling-based method. The conclusion drawn from the benchmark evaluation aligns with that concluded from the language modeling loss.

Table 3: Experimental results under different number of frames, with the sampling-based method.

| # Frames | # Embed./ Frames | Event-Bench | VNBench | MLVU | LongVideo Bench | VideoMME wo/w-subs | Avg. |
|---|---|---|---|---|---|---|---|
| 8 | 49 | 21.67 | 15.70 | 46.30 | 44.73 | 44.85/52.74 | 37.67 |
| 16 | 49 | 22.67 | 23.33 | 52.53 | 46.78 | 49.74/57.59 | 42.11 |
| 32 | 49 | 21.33 | 29.93 | 54.84 | 47.16 | 49.96/58.37 | 43.60 |
| 48 | 49 | 22.67 | 34.15 | 56.22 | 48.75 | 52.81/59.11 | 45.62 |
| 64 | 49 | 25.33 | 32.59 | 57.23 | 47.08 | 52.59/58.93 | 45.63 |
| 96 | 49 | **26.67** | 37.26 | 60.97 | 48.60 | 53.26/60.85 | 47.94 |
| 128 | 49 | 25.67 | **39.70** | **61.44** | **51.40** | **56.11/61.63** | **49.33** |

---

**Take-away Findings**

- Increasing the number of visual embeddings can significantly enhance the performance, with the sampling-based method achieves the peak at 49 tokens while the compression-based method does not saturate even with 196 tokens.

- When the visual context window size is limited, the compression-based method can effectively preserve more visual information with fewer visual embeddings.

---

## 3.3 SCALING EFFECT OF THE SELECTED FRAMES

Next, we continue to explore the scaling effect of the selected frames by varying its number $T$ while fixing the number of embeddings per frame $M$. We also consider utilizing sampling-based and compression-based methods.

### 3.3.1 SAMPLING-BASED METHOD

**Experimental Setup.** In existing works, it has become a widely used practice to sample frames uniformly from the original video to accommodate for the context length of LLM. Based on this method, we sample different numbers of frames from the video to explore the scaling effect, by varying $T$ in $\{1, 8, 16, 32, 48, 64, 96, 128\}$. In Section 4, we further increase $T$ to 162 to explore the limit of scaling frames. The maximum context length allowed by the computation memory is 8K, corresponding to 128 frames and 49 visual embeddings per frame. As a result, we adopt $4 \times 4$ MeanPooling to keep 49 embeddings per frame and train 8 video MLLMs by varying the number of frames from 1 to 128.

**Fitting Function.** Similarly, we use the following function to fit the scaling law of frames:

$$\mathcal{L}(T) = L_T + \left(\frac{T_0}{T}\right)^{a_T} \tag{4}$$

We fit the losses with the number of frames $T$ and obtain $L_T = 0.14, T_0 = 5.37 \times 10^{-7}, \alpha_T = 0.04$, with $R^2 = 0.892$. The fitted curve in Figure 2a shows that $\mathcal{L}(T)$ decreases with increasing $T$, following a power-law like trend. We calculate the mean squared error between the actual loss and predicted loss, obtaining a value of 0.0001, which indicates a very low fitting error.

**Benchmark Performance Analysis.** The results in Table 3 and Figure 3b show that as the number of frames increases, the model consistently improves on all benchmarks, with no clear saturation point, even at 128 frames, which exceeds the maximum frame count of most video MLLMs. Among all the benchmarks, VNBench shows that the most pronounced improvements (from 15.70 to 39.70), suggesting that the Needle-In-the-Haystack-Search (NIAH) task benefits most from extended temporal context. However, the Event-Bench shows no significant improvement beyond 64 frames, and a detailed inspection reveals that all questions in Event-Bench focus on episodic reasoning, a task that cannot be effectively learned by video MLLMs simply by increasing the number of frames (Li

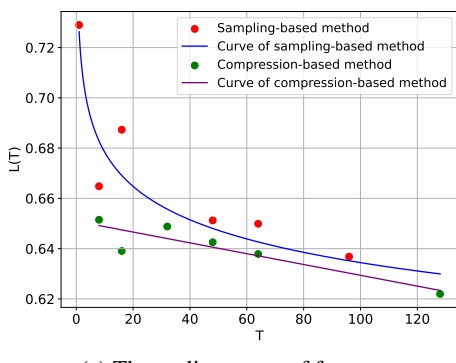

(a) The scaling curve of frames.

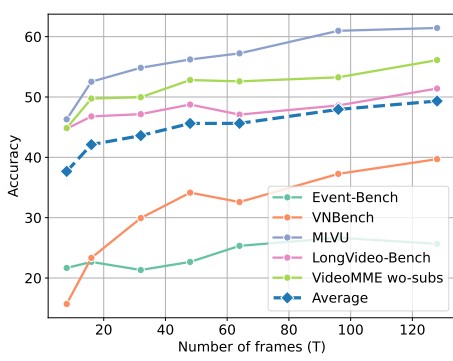

(b) The relationship between the number of frames and the benchmark accuracy.

Figure 3: The scaling law of frames, reflected by the language modeling loss and the zero-shot accuracy on video understanding benchmarks.

Table 4: Model performance with the compression-based method under different numbers of frames. For the model trained with 128 frames, since $T_{max} = T = 128$ in this setting, the temporal pooling kernel size is $l = \lceil \frac{128}{128} \rceil = 1$, resulting in the same outcome as the sampling-based method.

| # Frames | # Embed./ Frames | Event- Bench | VNBench | MLVU | LongVideo Bench | VideoMME wo/w-subs | Avg. |
|---|---|---|---|---|---|---|---|
| 8 | 49 | 25.67 | 20.89 | 53.04 | 46.32 | 50.48/57.41 | 42.30 |
| 16 | 49 | 27.00 | 27.04 | 55.77 | 48.07 | 50.44/58.30 | 44.44 |
| 32 | 49 | 25.33 | 29.78 | 59.37 | 48.52 | 53.81/60.93 | 46.29 |
| 48 | 49 | 24.33 | 37.41 | 59.31 | 47.61 | 52.07/59.81 | 46.76 |
| 64 | 49 | 29.00 | 36.30 | 61.03 | 47.61 | 53.70/60.56 | 48.03 |
| 128* | 49 | 25.67 | 39.70 | 61.44 | 51.40 | 56.11/61.63 | 49.33 |

et al., 2023d). Overall, compared to scaling the visual embeddings per frame (Figure 2b), increasing the frames is more beneficial for improving the model performance.

**Performance Compensating for Compressing Visual Embeddings.** Another interesting finding is that the performance degradation caused by compressing visual embeddings can be compensated by increasing the number of frames. Specifically, Table 2 shows that reducing the number of visual embeddings per frame from 196 to 49 leads to a performance drop across all benchmarks. However, if we simultaneously increase the number of frames to 128, the accuracy returns, even surpassing the model with 196 embeddings (comparing the last rows of Table 2 and Table 3, both setups use a total of 6272 visual embeddings, but one utilizes 32 frames with 196 embeddings per frame, while the other employs 128 frames with 49 embeddings per frame). These results suggest that when constrained by the visual context length, we can increase the number of frames while decreasing the embeddings per frame to achieve better performance, as will be further demonstrated in Section 4.

### 3.3.2 COMPRESSION-BASED METHOD

**Experimental Setup.** Compressing frames along the temporal dimension has been widely discussed in the field of video representation learning but remains underexplored in video MLLMs (Cheng et al., 2024). Similar to the compression strategy used in Section 3.2, we utilize MeanPooling here to reduce the number of frames input to the LLM. Specifically, we uniformly sample $T_{max}$ frames[1] from the video and encode them with the image encoder. Then, we apply MeanPooling along the temporal dimension to compress the video into $T$ frames, where the temporal pooling kernel size $l$ is determined by $T_{max}$ and $T$: $l = \lceil \frac{T_{max}}{T} \rceil$. Due to the limitation of computational memory, we set $T_{max} = 128$ and $T = \{8, 16, 32, 48, 64, 128\}$ to explore the scal-

---

[1] If the original video duration $T' \leq T_{max}$, we uniformly sample $T'$ frames from it; otherwise, we uniformly sample $T_{max}$ frames, which is a common practice for video MLLMs.

ing law, which is significantly larger than existing state-of-the-art video MLLMs that mostly use 32 or 64 frames as input (Li et al., 2024a; Cheng et al., 2024). To ensure a fair comparison with the sampling-based method, we also reduce the number of visual embeddings per frame to 49. In practice, we utilize three-dimensional MeanPooling, instead of first performing spatial MeanPooling followed by temporal MeanPooling, to avoid over-smoothing of the feature maps.

**Fitting Function.** Different from the previous experiments, a power-law like function can't fit the data points in this part. Instead, we find that a simple linear function can well describe the function relationship, which is defined as follows:

$$\mathcal{L}(T) = a \times T + b \tag{5}$$

We fit the losses with the number of frames $T$ and obtain $a = -0.0002, b = 0.651$, with $R^2 = 0.807$. The fit curve is shown in Figure 3a. We calculate the mean squared error between the actual loss and predicted loss is $1.753 \times 10^{-5}$, which indicates a very low fitting error.

**Benchmark Performance Analysis.** Comparing the curve of the sampling-based and the compression-based methods, the latter consistently results in lower loss. This phenomenon reveals the temporal redundancy in video data, showing that temporal information can be effectively preserved even when compressed into fewer frames. The evaluation results on the benchmarks are shown in Table 4. Overall, increasing the number of frames consistently improves model performance. Furthermore, the compression-based method achieves higher accuracy than the sampling-based method for the same number of frames. This aligns with the phenomenon that the compression-based method generally gives lower training loss as depicted in Figure 3a.

> **Take-away Findings**
>
> - Increasing the number of frames consistently improves the performance, even compensating for the performance degradation caused by compressing visual embeddings per frame.
>
> - When the visual context window size is limited, the compression-based method can preserve more temporal information than sampling-based method with fewer frames.

To validate the reliability of our scaling law, we perform predictive validation by holding out the last data point, refitting the scaling function using the remaining points, and then predicting the loss for the hold-out point. As shown in Table 14 in the E.2, the predicted losses align closely with the actual losses, with prediction errors below 0.01. This validates the reliability of our scaling law and ensures the robustness of our findings.

## 4 TRADE-OFF BETWEEN VISUAL EMBEDDINGS AND FRAMES

Section 3.2 and Section 3.3 have discussed the scaling effect of visual embeddings and frames separately. In this section, we explore the joint effect of the two factors, and study the problem: *how to jointly determine the numbers of visual embeddings and frames under the constrain of maximum input length of LLM or deployment resource?*

**Fitting Function of the Two Factors.** Following Hoffmann et al. (2022), we fit the losses by considering the numbers of embeddings $M$ and frames $T$ as follows:

$$\mathcal{L}(M,T) = C_M \times M^{-\alpha} + C_T \times T^{-\beta} + L_0 \tag{6}$$

Specifically, we set the number of visual embeddings as $\{25, 81\}$, and set the number of frames as $\{48, 64, 80, 96\}$, train $2 \times 4 = 8$ models in total. To extend the data points, we also include the 17 models trained in Section 3, and finally obtain 25 models in total. We obtain $C_M = 0.25, \alpha = 0.26, C_T = 0.13, \beta = 0.21, L_0 = 0.50$, with $R^2 = 0.884$. The fit curve along the axes of $T$ and $M$ is shown in Figure 4. With the decreasing of both $T$ and $M$, the loss $\mathcal{L}(M,T)$ will consistently increase, reaching the highest loss at the data point $T = 32, M = 4$ in our experiment. In contrast,

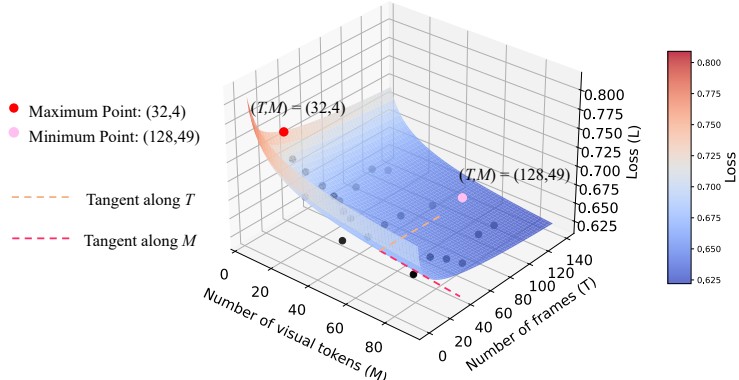

Figure 4: The scaling law of the visual embeddings and frames. We also display the maximum point and the minimum point in our experiments.

Table 5: Model performance under the visual context length of 6K.

| # Frames | # Embed./ Frames | # Total Embed. | Loss ↓ | Event-Bench | VNBench | MLVU | LongVideo Bench | VideoMME wo/w-subs | Avg. |
|---|---|---|---|---|---|---|---|---|---|
| 8 | 729 | 5832 | 0.78 | 18.33 | 16.30 | 50.98 | 43.44 | 46.22/53.67 | 38.16 |
| 30 | 196 | 5880 | 0.71 | 28.67 | 31.11 | 54.97 | 48.90 | 53.19/60.19 | 46.17 |
| 72 | 81 | 5832 | 0.70 | 24.33 | 37.56 | 58.37 | **50.34** | 53.04/61.11 | 47.46 |
| 120 | 49 | 5880 | **0.68** | 29.67 | 38.44 | 59.06 | 49.81 | 55.15/61.67 | 48.97 |
| 162 | 36 | 5832 | 0.70 | **33.00** | **40.67** | **62.83** | 50.04 | **55.19/62.00** | **50.62** |

as the $M$ and $T$ increase, the loss gradually decreases, and $T = 128, M = 49$ reaches the lowest loss. The computed gradient via the fitting function can help determine whether to increase $M$ or $T$ to achieve a lower loss. For example, the derivatives at $T = 32, M = 4$ are $\frac{\partial \mathcal{L}}{\partial M} = -0.01, \frac{\partial \mathcal{L}}{\partial T} = -0.004$, indicating that $\mathcal{L}$ descends faster along the $M$ direction. Therefore, increasing the number of embeddings is more promising to obtain lower loss, which aligns with our experiments.

**Finding Optimal Setting.** In practice, we are interested in the question that "given the visual context window $L$, what is the best choice of $M$ and $T$ that achieves the lowest loss $\mathcal{L}(M,T)$"? To answer this question, we utilize the Lagrange multiplier method to obtain the minimum point of Equation 6 under the constraint $M \times T < L$:

$$T_{\text{opt}} = \left( \frac{L}{\left( \frac{\beta C_T}{\alpha C_M} \right)^{\frac{1}{1-\alpha}}} \right)^{\frac{1-\alpha}{2-\beta-\alpha}}, \qquad M_{\text{opt}} = \left( \frac{\beta C_T}{\alpha C_M} \right)^{\frac{1}{1-\alpha}} T^{\frac{1-\beta}{1-\alpha}} \tag{7}$$

To verify the effectiveness of this principle, we set $L$ as 6K and obtain $\langle T_{\text{opt}}, M_{\text{opt}} \rangle \approx \langle 118, 51 \rangle$ according to Equation 7. For the experiment, we vary the number of visual embeddings and frames simultaneously under a fixed visual context length, yielding five $\langle T, M \rangle$ configurations: $\langle 8, 729 \rangle, \langle 30, 196 \rangle, \langle 72, 81 \rangle, \langle 120, 49 \rangle, \langle 162, 36 \rangle$. We then train five video MLLMs based on these configurations. The results in Table 5 show that the minimum loss is achieved with 120 frames and 49 visual embeddings per frame, which is quite near to $\langle T_{\text{opt}}, M_{\text{opt}} \rangle \approx \langle 118, 51 \rangle$. As for the benchmark evaluation, scaling the number of frames consistently improved overall performance without saturation, even with 162 frames and 36 visual embeddings. This phenomenon occurs because there remains a gap between the next-token-prediction loss and final performance on downstream tasks, but the theoretical minimum point can serve as a strong starting point for subsequent optimization. To verify the generalization of our conclusion, we conduct experiments on other vision encoders and LLM backbones in Appendix B. We also derive the $T_{\text{opt}}$ and $M_{\text{opt}}$ for MLLMs with different context lengths, ranging from 4K to 126K, and verify the conclusion in Appendix C.

## 5 RELATED WORK

**Scaling Law.**   In the field of LLMs, scaling model parameters and training data have been shown to consistently enhance model capacity (Radford et al., 2019; Brown, 2020; Touvron et al., 2023). As a result, it is necessary to build a quantitative relationship between these scaling factors and the final performance, which is called the scaling law. Two representative scaling laws for LLM are proposed by Kaplan et al. (2020) and Hoffmann et al. (2022), where the former one models the relationship between the loss and model size, dataset size, and the amount of computation budget independently, and the follower one models the relationship between loss and model size, dataset size jointly. Inspired by these works, subsequent studies show that the scaling law also holds for different model architectures (Clark et al., 2022), training strategies (Gao et al., 2023), and can be transferred to other domains like computer vision (Zhai et al., 2022; Dehghani et al., 2023) and multi-modal (Radford et al., 2021; Alayrac et al., 2022).

**Video MLLM.**   Training an MLLM Liang et al. (2024); Yao et al. (2024b); Du et al. (2023) with long video understanding ability is a challenging task and remains underexplored. One line of work focuses on enabling long video training from the perspectives of training systems (Xue et al., 2024), training strategies (Liu et al., 2024b), and model architectures (Wang et al., 2024b). For example, LongVILA (Xue et al., 2024) proposes the first Multi-Modal Sequence Parallelism system for long-context training and inference. Kangaroo (Liu et al., 2024b) utilizes a curriculum training pipeline to gradually increase the number of frames during training. LongLLaVA (Wang et al., 2024b) adapts the model architecture to a hybrid of Mamba (Gu & Dao, 2023) and Transformer (Vaswani et al., 2017) blocks. Another line of work aims to enable long video understanding during inference (Song et al., 2024; Zhang et al., 2024a). For example, MovieChat (Song et al., 2024) proposes a memory mechanism that includes a rapidly updated short-term memory and a compact long-term memory to store representations of long videos. LongVA (Zhang et al., 2024a) extends the context window of an LLM and demonstrates that long video understanding can be directly transferred from an MLLM without any video-specific training.

**Visual Embedding Compression.**   Most MLLMs consist of a vision encoder, an LLM, and a visual projector to project the image embeddings into the semantic space of the LLM. Early works like Flamingo (Alayrac et al., 2022) adopt a resampler, which inserts a cross-attention module into the LLM layer to extract visual features, and this is followed by IDEFICS (Laurençon et al., 2024a) and Otter (Li et al., 2023a). Similarly, BLIP-2 (Li et al., 2023b) and InstructBLIP (Dai et al., 2023) utilize a cross-attention module called Q-Former to compress the image embeddings before inputting them into the LLM. Another line of work, represented by LLaVA (Liu et al., 2024a), directly projects the image embeddings into the semantic space of the LLM with an MLP, achieving decent performance and converging quickly. Extensions like (Yao et al., 2024a; Cai et al., 2024) add pooling modules after the MLP to reduce visual embeddings. However, projector design for video tasks is less explored. LLaVA-NeXT-Video (Zhang et al., 2024c) and LLaVA-OneVision (Li et al., 2024a) use mean pooling or bilinear interpolation to aggregate visual embeddings, while neglecting the temporal dependency of video frames. To model the temporal dependency, VideoLLaMA2 (Cheng et al., 2024) introduces a downsampling module and a spatial-temporal convolution module.

## 6 CONCLUSION

In this work, we explored the basic design space of visual context representation in video MLLMs from two major aspects: the number of frames per video (*frame selection*) and visual embeddings per frame (*embedding selection*). Using a widely-used Video-MLLM architecture, we tested various sampling and compression strategies, collecting data points by varying frames and embeddings. We formulated the studied task as a constrained optimization problem, and then studied the scaling effects for frame selection and embedding selection. Then we fitted the performance function curve *w.r.t.* the two factors, and derived several important empirical findings to determine the two factors. Finally, we modeled their joint effects, derived the optimal setting, and validated the effectiveness with empirical experiments. Our findings highlight the significant impact of visual context representation on video MLLM performance, which is worth more research attention. Future work will explore advanced strategies for frame and embedding selection and develop architectures better suited for long video representation.

ACKNOWLEDGEMENTS

This work was partially supported by National Natural Science Foundation of China under Grant No. 92470205 and 62222215, Beijing Municipal Science and Technology Project under Grant No. Z231100010323009, and Beijing Natural Science Foundation under Grant No. L233008. This research was also supported by Baichuan and the Outstanding Innovative Talents Cultivation Funded Programs 2024 of Renmin University of China. Xin Zhao and Bingning Wang are the corresponding authors.

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

# A  IMPLEMENTATION DETAILS

We list all the training data we used in Table 6, and the training hyperparameters are in Table 7.

Table 6: The statistics of our training data, including 1.8M image-text instructions and 0.7M video-text instructions.

| Modality | Dataset | Samples |
|---|---|---|
| Image-Text | Cauldron | 1.8M |
| Video-Text | VideoChatGPT-100K | 100K |
| | ShareGPT4Video | 40K |
| | ShareGPTVideo | 255K |
| | VIM | 32K |
| | NExT-QA | 40K |
| | SthSthV2 | 40K |
| | STAR | 40K |
| | TextVR | 40K |
| | CLEVRER | 80K |
| | Kinetics-710 | 40K |
| Total | - | 2.5M |

Table 7: Training hyperparameter.

| Hyperparameter | Value |
|---|---|
| Global batch size | 64 |
| Gradient clipping | 1 |
| Weight decay | 0 |
| Warmup ratio | 0.03 |
| LLM lr | 2e-5 |
| Projector lr | 1e-4 |
| Vision encoder lr | 2e-6 |
| lr schedule | cosine |

# B  EXPERIMENTS ON OTHER BACKBONES

To verify the generalization of our conclusion in Section 4, we utilize other representative base models: LLaMA3-8B (Meta, 2024) and CLIP-ViT-L-336px (Radford et al., 2021). Experiments are conducted under the same settings as described in Section 4. The experimental results in Table 8 show that both the minimum loss and the optimal benchmark performance are achieved with 120 frames and 49 visual embeddings per frame, consistent with the conclusion in Section 4. This demonstrates the generalizability of our conclusion and are expected to inspire future developments in video MLLMs.

Table 8: Model performance under the same number of visual embeddings. The backbone models are LLaMA3-8B and CLIP-ViT-L-336px.

| # Frames | # Embed./ Frames | # Total Embed. | Loss ↓ | Event-Bench | VNBench | MLVU | LongVideo Bench | VideoMME wo/w-subs | Avg. |
|---|---|---|---|---|---|---|---|---|---|
| 10 | 576 | 5760 | 0.90 | 20.00 | 18.52 | 49.95 | 45.80 | 44.22/43.37 | 36.98 |
| 30 | 196 | 5880 | 0.91 | 23.33 | 26.15 | 52.98 | 45.80 | 45.74/46.41 | 40.07 |
| 72 | 81 | 5832 | 0.89 | 19.33 | 26.67 | 54.99 | 46.03 | 46.59/47.26 | 40.15 |
| 120 | 49 | 5880 | **0.85** | **24.33** | **30.67** | **57.68** | 47.16 | 46.93/47.89 | **42.44** |
| 162 | 36 | 5832 | 0.86 | 22.00 | 28.74 | 56.80 | **47.99** | **49.15/49.67** | 42.39 |

## C GENERALIZATION TO DIFFERENT VISUAL CONTEXT LENGTHS

Besides the optimal settings for 6K context window length, we also derive the $T_{\text{opt}}$ and $M_{\text{opt}}$ for MLLMs with different context lengths, ranging from 4K to 126K, in Table 9. To verify this conclusion, we conduct additional experiments with a visual context window length of 4K. The results in Table 10 show that the optimal performance is achieved with 96 frames and 36 visual embeddings per frame, which closely aligns with the predicted $\langle T, M \rangle = \langle 97, 41 \rangle$. Due to memory constraints, we leave the validation of longer context lengths as future work.

Table 9: Estimated optimal visual tokens and frames for various visual context lengths.

| Visual Context Length | # Visual Tokens ($M$) | # Frames ($T$) | Optimal Ratio ($\frac{T}{M}$) |
|---|---|---|---|
| 4,000 | 41 | 97 | 2.4 |
| 6,000 | 51 | 118 | 2.3 |
| 14,000 | 78 | 178 | 2.3 |
| 30,000 | 116 | 258 | 2.2 |
| 62,000 | 169 | 367 | 2.2 |
| 126,000 | 243 | 517 | 2.1 |

Table 10: Model performance under the visual context length of 4K.

| # Frames | # Embed./ Frames | # Total Embed. | Loss ↓ | Event-Bench | VNBench | MLVU | LongVideo Bench | VideoMME wo/w-subs | Avg. |
|---|---|---|---|---|---|---|---|---|---|
| 18 | 196 | 3528 | 0.8010 | 25.00 | 24.81 | 54.75 | 48.68 | 50.15/50.41 | 42.30 |
| 70 | 49 | 3430 | 0.7581 | 23.00 | 32.52 | 59.60 | 49.36 | **54.44/55.33** | 45.71 |
| 96 | 36 | 3456 | **0.7539** | **28.00** | 35.63 | **61.64** | **49.51** | 53.26/53.78 | **46.97** |
| 138 | 25 | 3450 | 0.7554 | 22.33 | **37.93** | 60.06 | 47.84 | 52.26/53.19 | 45.60 |

## D SCALING THE VISUAL EMBEDDINGS IN VISION ENCODER

In Section 3.2, we fix the number of frames and vary the number of visual embeddings input to the LLM to study the effect of scaling visual embeddings. Besides varying the visual embeddings input to the LLM, we can also vary the visual embeddings produced by the vision encoder. In this section, we explore which strategy contributes more to the performance improvements.

Generally speaking, two methods can increase the number of visual embeddings produced by a ViT-based vision encoder: (1) increasing the resolution of the frame, or (2) increasing the number of patches in each frame. To explore the contributions of longer visual embeddings after the vision encoder versus longer visual tokens input to the LLM, we conducted two sets of experiments based on these two methods.

**Increasing the resolution of the frames.** To increase the encoded resolution of the frames while keeping other factors the same, we utilize SigLIP-base-patch16-256 and SigLIP-base-patch16-512 as the vision encoders and design the following three experiments:

- Exp1 (Baseline): Standard setup with 256 visual tokens after the vision encoder and 16 tokens (setting kernel size as 2) input to the LLM.
- Exp2: Increasing the number of tokens after the vision encoder to 1024 while keeping the number of tokens input to the LLM constant at 16.
- Exp3: Keeping the number of tokens after the vision encoder at 256 while increasing the number of tokens input to the LLM to 64.

From the results in Table 11, we observe: (1) Exp3 consistently improves performance across all benchmarks, demonstrating the benefit of increasing the number of visual tokens input to the LLM. (2) Exp2 shows slight performance degradation on some benchmarks compared to Exp1, which is

Table 11: Comparing the influence of scaling visual embeddings in vision encoder versus in LLM. We increase the resolution of the frames to produce more visual embeddings in vision encoder.

| | # Emb. in VE | # Emb. for LLM | Event-Bench | VNBench | MLVU | LongVideo Bench | VideoMME wo/w-subs | Avg. |
|---|---|---|---|---|---|---|---|---|
| Exp1 | 256 | 16 | 19.33 | 25.26 | 51.98 | 45.04 | 47.93/48.19 | 39.62 |
| Exp2 | 1024 | 16 | 23.67 | 24.89 | 53.74 | 44.97 | 47.70/48.78 | 40.63 |
| Exp3 | 256 | 64 | 20.33 | 26.96 | 54.04 | 46.56 | 48.59/49.48 | 40.99 |

Table 12: Comparing the influence of scaling visual embeddings in vision encoder versus in LLM. We increase the number of patches in each frame to produce more visual embeddings in the vision encoder.

| | # Emb. in VE | # Emb. for LLM | Event-Bench | VNBench | MLVU | LongVideo Bench | VideoMME wo/w-subs | Avg. |
|---|---|---|---|---|---|---|---|---|
| Exp1 | 49 | 16 | 25.0 | 24.67 | 50.65 | 45.87 | 44.22/46.00 | 39.40 |
| Exp2 | 196 | 16 | 13.33 | 22.22 | 47.83 | 40.12 | 41.85/40.93 | 34.38 |
| Exp3 | 49 | 49 | 22.0 | 24.81 | 49.84 | 47.16 | 47.15/46.19 | 39.53 |

expected since more tokens after the vision encoder introduce greater information loss during token compression/sampling to fit within the fixed LLM input size.

**Increasing the number of patches per frame.** To increase the number of patches per frame while keeping other factors the same, we utilize clip-base-patch32-224 and clip-base-patch16-224 as the vision encoders and design the following three experiments:

- Exp1 (Baseline): Standard setup with 49 tokens after the vision encoder and 16 tokens (setting kernel size as 2) input to the LLM.

- Exp2: Increasing the number of tokens after the vision encoder to 196 while keeping the number of tokens input to the LLM constant at 16.

- Exp3: Keeping the number of tokens after the vision encoder at 49 while increasing the number of tokens input to the LLM to 49.

From the results in Table 12, we observe: (1) Exp2 shows consistent performance degradation across all benchmarks. The reason is the same as in the above experiment: more visual tokens after the vision encoder means that we may lose more information during token sampling/compression in order to keep the number of visual tokens input to LLM the same. (2) Exp3, which increases the number of tokens input to the LLM, improves the performance on most benchmarks, further emphasizing the benefits of directly optimizing the LLM input size.

The results from both sets of experiments consistently show that increasing the number of visual tokens input to the LLM is significantly more effective than increasing the number of visual tokens after the vision encoder. The findings highlight the importance of optimizing the LLM input for improved performance, as excessive compression of tokens after the vision encoder leads to information loss that cannot be recovered downstream.

# E    VERIFICATION OF THE SCALING EXPERIMENTS

## E.1    ROBUSTNESS VALIDATION WITH MULTIPLE SEEDS

To confirm the consistency of our results, we trained the model with a widely used configuration $\langle T, M \rangle = \langle 32, 196 \rangle$ using five different random seeds. The results in Table 13 demonstrate minimal variability in both loss and accuracy, with a particularly low standard deviation for the loss (0.0002). This confirms that our training process is robust and not significantly affected by randomness. Furthermore, the fluctuations in loss are smaller than those in accuracy on downstream tasks. Therefore, we used the loss as a stable metric to estimate the parameters of the scaling law function.

Table 13: The evaluation results of models trained with different random seeds.

| Seed | Loss | ACC (%) |
|------|------|---------|
| 1 | 0.6377 | 47.20 |
| 2 | 0.6382 | 47.01 |
| 3 | 0.6377 | 46.62 |
| 4 | 0.6381 | 46.75 |
| 5 | 0.6379 | 47.11 |
| Avg. | 0.6379 | 46.94 |
| Std. | 0.0002 | 0.219 |

Table 14: The predicted loss according to the scaling function and the actual loss.

| Setup | Hold-out point | Predicted loss | Actual loss |
|-------|----------------|----------------|-------------|
| Sampling tokens | $M = 196$ | 0.678 | 0.679 |
| Compressing tokens | $M = 196$ | 0.630 | 0.638 |
| Sampling frames | $T = 128$ | 0.628 | 0.638 |
| Compressing frames | $T = 128$ | 0.629 | 0.622 |

## E.2 PREDICTIVE VALIDATION OF THE SCALING LAW

We conducted predictive validation by holding out the last data point, refitting the scaling function with the remaining points, and predicting the loss for the hold-out point. The predicted losses were then compared to the actual losses. As shown in Table 14, the predicted losses consistently align with the actual values, with prediction errors remaining below 0.01. These results validate the reliability of our scaling law and confirm the robustness of our findings.

## F COMPARION WITH EXISTING MLLMS

We compare our model with a series of representative MLLMs, including four proprietary MLLMs: GPT-4o (OpenAI, 2024), Gemini-1.5-Pro (Reid et al., 2024), GPT-4V (OpenAI, 2023), Qwen-VL-Max (Bai et al., 2023), as well as ten open-source MLLMs: Video-CCAM (Fei et al., 2024), LLaMA-VID-long (Li et al., 2023d), MovieChat (Song et al., 2024), ST-LLM (Liu et al., 2025), VideoLLaMA2 (Cheng et al., 2024), LongVA (Zhang et al., 2024a), and LongViLA (Xue et al., 2024), LLaVA-OneVision (Li et al., 2024a), LLaVA-Video (Zhang et al., 2024d), and Qwen2-VL (Wang et al., 2024a). We evaluate our model and the baseline models on the long video understanding benchmarks. The results in Table 15 show that there is still a large gap between our model and SOTA results, primarily due to differences in training data. To mitigate this gap and demonstrate the superiority of our visual context representation scheme, we incorporate the recently proposed video instruction collection, Video178K (Zhang et al., 2024d), to replace the instructions used in our original experiments. Under this configuration, we re-train the model with $\langle T, M \rangle = \langle 120, 49 \rangle$ (denoted as "Ours+Video178K"). This adjustment yielded substantial performance improvements: our results on VNBench and LongVideoBench surpassed the SOTA, while Video-MME became comparable to the SOTA. However, MLVU and Event-Bench remain below SOTA levels. The disparity on MLVU and Event-Bench can be attributed to the fact that the model in Zhang et al. (2024d) is built upon a powerful image MLLM pre-trained on approximately 8M image instructions, a scale not yet reflected in our experimental setup. Nonetheless, these results underscore the effectiveness of our proposed experimental recipes and their potential when paired with richer data. In the future, we plan to incorporate more diverse and extensive image instructions to further enhance our video MLLM's capabilities and close the remaining gaps with SOTA. Notably, our model even outperforms GPT-4V on certain benchmarks, such as Event-Bench and MLVU, demonstrating the effectiveness of our optimal visual context representation scheme.

Table 15: Experiment results on representative long video understanding benchmarks.

| Models | Training Data | Event-Bench | VNBench | VideoMME wo/w-subs | MLVU | LongVideoBench |
|---|---|---|---|---|---|---|
| *Proprietary MLLMs* | | | | | | |
| GPT-4o | Unknown | 37.33 | 64.4 | 71.9/77.2 | **64.6** | **66.7** |
| Gemini-1.5-Pro | Unknown | **38.67** | **66.7** | **75.0/81.3** | - | 64.0 |
| GPT-4V | Unknown | 27.00 | 48.9 | 59.9/63.3 | 49.2 | 59.1 |
| Qwen-VL-Max | Unknown | - | - | 51.3/51.2 | 42.2 | - |
| *Open-source MLLMs* | | | | | | |
| Video-CCAM-14B | 4.4M | - | - | 53.2/57.4 | 63.1 | - |
| LLaMA-VID-long-7B | 1.6M | 0.00 | 10.8 | - | 33.2 | - |
| MovieChat-7B | Unknown | 20.33 | - | - | 25.8 | - |
| ST-LLM-7B | Unknown | 16.67 | 22.7 | 37.9/42.3 | - | - |
| VideoLLaMA2-7B | 13.4M | - | - | 46.6/- | 48.5 | - |
| LongVA-7B | 1.3M | - | - | 52.6/- | 56.3 | - |
| LongViLA-8B | Unknown | - | - | 50.5/- | - | - |
| Qwen2-VL-7B | Unknown | - | 33.9 | **63.3**/69.0 | - | 56.8 |
| LLaVA-OneVision-7B | 7.7M | 28.7 | 35.7 | 58.2/61.5 | 64.7 | 56.5 |
| LLaVA-Video-7B | 9.3M | **43.0** | 40.7 | **63.3/69.7** | **66.9** | 58.2 |
| Ours | 2.6M | 29.7 | 38.4 | 55.2/61.7 | 59.1 | 49.8 |
| Ours+Video178K | 3.4M | 35.0 | **67.3** | 61.7/66.9 | 66.0 | **58.4** |

## G EXPERIMENTAL RESULTS WITHOUT ANY FRAMES AS INPUT

We also report the results without any frames as the input. The results in Table 16 show that: (1) Event-Bench and VNBench are the most reliant on visual input. (2) Comparing the results with Section 3, the model achieves substantial gains across all benchmarks when visual inputs are included. These findings highlight the critical role of visual tokens in enabling video MLLMs to achieve high performance on diverse benchmarks.

Table 16: Evaluation results without any frames as the input.

| # Frames | Event-Bench | VNBench | MLVU | LongVideo Bench | VideoMME wo/w-subs | Avg. |
|---|---|---|---|---|---|---|
| 0 | 19.30 | 7.56 | 40.48 | 36.64 | 34.89/36.48 | 29.23 |

