# OpenReview forum: "Exploring the Design Space of Visual Context Representation in Video MLLMs"
_ICLR.cc/2025/Conference — ICLR 2025 Poster_

### Official Review · Reviewer_WRue · 2024-11-02

**Soundness:** 3
**Presentation:** 3
**Contribution:** 2
**Rating:** 6
**Confidence:** 3

**Summary:**

This paper analyses the input design space of Video Multimodal Large Language Models (MLLMs) by scaling two components: the number of frames and the number of tokens per frame.
Given the quadratic computational complexity of the attention mechanism in the number of input tokens, MLLMs are limited in the size of their input. This paper studies the impact of varying the number of frames and the number of tokens per frame, as well as the sampling and compression strategies to obtain the final tokens.

**Strengths:**

The paper offers a valuable analysis of visual context representation design in video MLLMs. The approach is straightforward, with clearly explained experimental settings and well-presented results, and the conclusions are well-supported by experiments.

It is interesting to see the contribution of mean pooling versus uniform sampling, both for selecting frames and selecting tokens.

The authors promise to release code and data for reproducing their results.

**Weaknesses:**

Comparison to SOTA results on the benchmarks.
It is understood that the goal of this paper is not to obtain SOTA results on the evaluated benchmarks, but rather to perform an analysis on the design space of visual context representation in video MLLMs. Nevertheless, it would be informative to the reader to have the information of how far the experimental recipes presented in this paper are from the SOTA.

**Questions:**

Figure 2b and 3b: In addition to the performance curves on each benchmark, it would be valuable to also plot the average over all of them.

Section 3: At the other end of the scaling spectrum, it would be interesting to also report the performance without any visual token at all (M=0) to see what would be the comparative performance of a "blind" MLLM.

Minor typo:
L325: "increasing the frames"

---

> ### Author Response · Authors · 2024-11-22
> **Official Response to Reviewer WRue**
>
> Thank you for your insightful suggestions. We have outlined our responses to your concerns below. If you have any additional questions, please feel free to let us know. We will do our best to address them.
>
> > **Weakness**: Comparison to SOTA results on the benchmarks. It is understood that the goal of this paper is not to obtain SOTA results on the evaluated benchmarks, but rather to perform an analysis on the design space of visual context representation in video MLLMs. Nevertheless, it would be informative to the reader to have the information of how far the experimental recipes presented in this paper are from the SOTA.
>
> **R1**: We follow the reviewer's suggestion and compare our result (achieved with 120 frames and 49 visual embeddings per frame) to the SOTA results on the benchmarks.
> | Model               | Video-MME | MLVU  | LongVideoBench | VNBench | Event-Bench |
> |:--------------------|:---------:|:-----:|:--------------:|:-------:|:-----------:|
> | 7B models SOTA      | **63.3** [1]  | **70.8** [2] | 58.2 [2]      | 61.6 [3] | **44.3** [2]    |
> | Ours                | 55.2      | 59.1   | 49.8           | 38.4    | 29.7        |
> | Ours+Video178K      | 61.7      | 66.0   | **58.4**           | **67.3**    | 35.0        |
>
> From the comparison, it is evident that our results initially show a significant gap from the SOTA, primarily due to differences in training data. To address this, we incorporated Video178K, a subset of training data in [2], to replace the instructions used in our original experiments (denoted as Ours+Video178K). This adjustment yielded substantial performance improvements: our results on VNBench and LongVideoBench surpassed the SOTA, while Video-MME became comparable to the SOTA. However, the performances on MLVU and Event-Bench remain below SOTA levels. The disparity on MLVU and Event-Bench can be attributed to the fact that the model in [2] is built upon a powerful image MLLM pre-trained on approximately 8M image instructions and mixed 3.2M image instructions during video instruction tuning, which is a scale not yet reflected in our experimental setup. Nonetheless, these results underscore the effectiveness of our proposed experimental recipes and their potential when paired with richer data. In the future, we plan to incorporate more diverse and extensive image instructions to further enhance our video MLLM's capabilities and close the remaining gaps with SOTA.
>
> [1] Qwen2-VL: Enhancing Vision-Language Model’s Perception of the World at Any Resolution
>
> [2] Video Instruction Tuning With Synthetic Data
>
> [3] Video-XL: Extra-Long Vision Language Model for Hour-Scale Video Understanding
>
> > **Q1**: Figure 2b and 3b: In addition to the performance curves on each benchmark, it would be valuable to also plot the average over all of them.
>
> **R1**: Thanks for the suggestion and we have plotted the average performance in the revised version.
>
> > **Q2**: Section 3: At the other end of the scaling spectrum, it would be interesting to also report the performance without any visual token at all (M=0) to see what would be the comparative performance of a "blind" MLLM.
>
> **R2**: Following the reviewer's suggestion, we trained a "blind" MLLM and evaluate it on the benchmarks without any visual tokens as the input. The results are summarized below:
>
> | # Frames | Event-Bench | VNBench | MLVU  | LongVideoBench | VideoMME wo/w-subs | Avg.  |
> |:--------:|:-----------:|:-------:|:-----:|:--------------:|:-----------------:|:-----:|
> |    0     |    19.30    |   7.56  |  40.48 |      36.64      |     34.89/36.48     | 29.23 |
>
> From these results, we draw the following conclusions:
>
> - Event-Bench and VNBench are the most reliant on the visual input.
>
> - Comparing the results in the above with Section 3, the model achieves substantial gains across all benchmarks when visual inputs are included.
>
> These findings highlight the critical role of visual tokens in enabling video MLLMs to achieve high performance on diverse benchmarks.
>
> **Q3**: Minor typo: L325: "increasing the frames"
>
> **R3**: Thanks for your carefully readining and we will check the writing of the whole paper.

---

> ### Author Response · Authors · 2024-11-25
> **Kindly Reminder for the Discussion**
>
> Dear Reviewer WRue,
>
> Thanks for your careful reading of our paper. We have made several improvements to our paper based on your feedback. Since the rebuttal stage is coming to an end, we are more than happy to hear your comments and address any of your further concerns during the remaining time.
>
> Best,
>
> Authors

---

> ### Author Response · Authors · 2024-11-28
>
> Dear Reviewer WRue,
>
> We sincerely appreciate the time and effort you have dedicated to reviewing our paper. In response to your comments, we have carefully conducted additional experiments and revised our paper accordingly:
>
> 1. We conducted additional experiments to compare our results with the SOTA.
>
> 2. We included the results of a "blind" video MLLM without frames as input.
>
> 3. We made detailed revisions to the manuscript based on your other recommendations.
>
> We would be grateful if you could let us know whether our revisions and responses address your concerns, or if there are additional questions or clarifications you would like us to provide. Since the rebuttal stage is coming to an end, we are more than happy to hear your comments and address any of your further concerns during the remaining time.
>
> Best,
>
> Authors

---

### Official Review · Reviewer_4WaN · 2024-11-04

**Soundness:** 3
**Presentation:** 3
**Contribution:** 3
**Rating:** 8
**Confidence:** 4

**Summary:**

Given a fixed Video-LLM window size, this paper examines the optimal selection of the number of video frames, the number of video tokens, and the compression/sampling methods used for frames and embeddings. The authors include experiments on both loss and benchmark evaluations.

**Strengths:**

1. The paper is clearly written and follows a logical structure.

2. The paper includes extensive experiments and offers a thorough analysis. Notably, the experiments that consider fixed frames, fixed embeddings per frame, and the trade-off between embeddings and frames are particularly useful for following research.

**Weaknesses:**

1. Is the design space for embedding and frame operations too narrow? The findings within this design space may be biased towards different LLM backbones and vision encoders. We suggest that the authors conduct experiments across a variety of LLM backbones and vision encoders to ascertain the generalizability of their findings. Or at least one more different LLM backbone and vision encoders.

2. In the experiments depicted in Figure 2 and Tables 1 & 2, the authors utilize the same T = 32 (L168) for all experiments, which results in video tokens of varying lengths within the same LLM context window size. It is generally assumed that longer tokens within a pretrained LLM context window size lead to better performance. How can it be determined whether longer visual embeddings after the vision encoder or longer visual tokens in the LLM contribute more significantly to performance improvements?

3. Since this paper aims to study the scaling effect, conducting a sufficient number of experiments with training models is crucial to mitigate the impact of randomness bias. The authors should clarify the number of models trained for each data point presented in experiments.

**Questions:**

My major concerns lie in the soundness of the experiments. Please find more details in the 'Weaknesses' section

---

> ### Author Response · Authors · 2024-11-22
> **Official Response to Reviewer 4WaN [1/3]**
>
> Thank you for your insightful suggestions. We have outlined our responses to your concerns below. If you have any additional questions, please feel free to let us know. We will do our best to address them.
>
> > **W1**: Is the design space for embedding and frame operations too narrow? The findings within this design space may be biased towards different LLM backbones and vision encoders. We suggest that the authors conduct experiments across a variety of LLM backbones and vision encoders to ascertain the generalizability of their findings. Or at least one more different LLM backbone and vision encoders.
>
> **R1**: Thanks for the insightful suggestion. We utilize other representative backbone models: **LLaMA3-8B** and **CLIP-ViT-L-336px**. Experiments are conducted under the same settings as described in Section 4. The experimental results in the following Table show that both the minimum loss and the optimal benchmark performance are achieved with **120** frames and **49** visual embeddings per frame, consistent with the conclusion in Section 4. This demonstrates the generalizability of our conclusion and are expected to inspire future developments in video MLLMs.
>
> | # Frames | # Embed./Frames | # Total Embed. | Loss ↓ | Event-Bench | VNBench | MLVU  | LongVideoBench | VideoMME wo/w-subs | Avg.   |
> |:----------:|:-----------------:|:----------------:|:--------:|:-------------:|:---------:|:-------:|:----------------:|:--------------------:|:--------:|
> | 10       | 576             | 5760           | 0.90   | 20.00       | 18.52   | 49.95 | 45.80          | 44.22/43.37        | 36.98  |
> | 30       | 196             | 5880           | 0.91   | *23.33*     | 26.15   | 52.98 | 45.80          | 45.74/46.41        | 40.07  |
> | 72       | 81              | 5832           | 0.89   | 19.33       | 26.67   | 54.99 | 46.03          | 46.59/*47.26*      | 40.15  |
> | 120      | 49              | 5880           | **0.85** | **24.33**   | **30.67** | **57.68** | *47.16*        | *46.93*/47.89      | **42.44** |
> | 162      | 36              | 5832           | *0.86* | 22.00       | *28.74* | *56.80* | **47.99**       | **49.15/49.67**    | *42.39* |

---

> ### Author Response · Authors · 2024-11-22
> **Official Response to Reviewer 4WaN [2/3]**
>
> > **W2**: In the experiments depicted in Figure 2 and Tables 1 & 2, the authors utilize the same T = 32 (L168) for all experiments, which results in video tokens of varying lengths within the same LLM context window size. It is generally assumed that longer tokens within a pretrained LLM context window size lead to better performance. How can it be determined whether longer visual embeddings after the vision encoder or longer visual tokens in the LLM contribute more significantly to performance improvements?
>
> Thanks for the insightful question. Actually, there are two methods that can increase the number of visual embeddings produced by a ViT-based vision encoder: (1) increasing the **resolution** of the frame, or (2) increasing the **number of patches** in each frame. To compare the values of longer visual embeddings after the vision encoder versus longer visual tokens input to the LLM, we conducted two sets of experiments based on these two methods.
>
> **(1) Increasing the resolution of the frames:**
>
> To increase the encoded resolution of the frames while keeping other factors the same, we utilize SigLIP-base-patch16-256 and SigLIP-base-patch16-512 as the vision encoders and design the following three experiments:
>
> - Exp1 (Baseline): Standard setup with 256 visual tokens after the vision encoder and 16 tokens (setting kernel size as 2) input to the LLM.
> - Exp2: Increasing the number of tokens after the vision encoder to 1024 while keeping the number of tokens input to the LLM constant at 16.
> - Exp3: Keeping the number of tokens after the vision encoder at 256 while increasing the number of tokens input to the LLM to 64.
>
> | Experiment | # Emb. in VE | # Emb. for LLM | Event-Bench | VNBench | MLVU  | LongVideoBench | VideoMME wo/w-subs | Avg.  |
> |:----------:|:------------:|:--------------:|:-----------:|:-------:|:-----:|:--------------:|:------------------:|:-----:|
> |    Exp1    |     256      |       16       |    19.33    |  25.26  | 51.98 |      45.04     |     47.93/48.19    | 39.62 |
> |    Exp2    |    1024      |       16       |    **23.67**    |  24.89  | 53.74 |      44.97     |     47.70/48.78    | 40.63 |
> |    Exp3    |     256      |       64       |    20.33    |  **26.96**  | **54.04** |      **46.56**     |    **48.59/49.48**    | **40.99** |
>
> From these results, we observe:
>
> - Exp3 consistently improves performance across all benchmarks, demonstrating the benefit of increasing the number of visual tokens input to the LLM.
> - Exp2 shows slight performance degradation on some benchmarks compared to Exp1, which is expected since more tokens after the vision encoder introduce greater information loss during token compression/sampling to fit within the fixed LLM input size.
>
> **(2) Increasing the number of patches per frame:**
>
> Here, we compare three setups:
> - Exp1 (Baseline): Standard setup with 49 tokens after the vision encoder and 16 tokens (setting kernel size as 2) input to the LLM.
> - Exp2: Increasing the number of tokens after the vision encoder to 196 while keeping the number of tokens input to the LLM constant at 16.
> - Exp3: Keeping the number of tokens after the vision encoder at 49 while increasing the number of tokens input to the LLM to 49.
>
> | Experiment | # Emb. in VE | # Emb. for LLM | Event-Bench | VNBench | MLVU  | LongVideoBench | VideoMME wo/w-subs | Avg.  |
> |:----------:|:------------:|:--------------:|:-----------:|:-------:|:-----:|:--------------:|:------------------:|:-----:|
> |    Exp1    |      49      |       16       |    **25.00**     |  24.67  | **50.65** |      45.87     |     44.22/46.00    | 39.40 |
> |    Exp2    |     196      |       16       |    13.33    |  22.22  | 47.83 |      40.12     |     41.85/40.93    | 34.38 |
> |    Exp3    |      49      |       49       |    22.00     |  **24.81**  | 49.84 |      **47.16**     |     **47.15/46.19**    | **39.53** |
>
> From these results, we observe:
> - Exp2 shows consistent performance degradation across all benchmarks. The reason is the same as in the above experiment: more visual tokens after the vision encoder means that we may lose more information during token sampling/compression in order to keep the number of visual tokens input to LLM the same.
> - Exp3, which increases the number of tokens input to the LLM, improves the performance on most benchmarks, further emphasizing the benefits of directly optimizing the LLM input size.
>
> The results from both sets of experiments consistently show that increasing the number of visual tokens input to the LLM is significantly more effective than increasing the number of visual tokens after the vision encoder. The findings highlight the importance of optimizing the LLM input for improved performance, as excessive compression of tokens after the vision encoder leads to information loss that cannot be recovered downstream.

---

> ### Author Response · Authors · 2024-11-22
> **Official Response to Reviewer 4WaN [3/3]**
>
> > **W3**: Since this paper aims to study the scaling effect, conducting a sufficient number of experiments with training models is crucial to mitigate the impact of randomness bias. The authors should clarify the number of models trained for each data point presented in experiments.
>
> **R3**: Thank you for pointing out the importance of conducting sufficient experiments to mitigate randomness bias. However, due to our limited computational resource, we can not train multiple models for every data point. Thus, before we start training, we take rigorous steps to validate the robustness of our experimental settings to test the reliability of our findings:
>
> **(1) Robustness Validation with Multiple Seeds**:
>
> To confirm the consistency of our results, we trained the model with a widely used configuration <T,M>=<32,196> using five different random seeds. The results are as follows:
>
> | Seed | Loss   | ACC (%) |
> |:----:|:------:|:-------:|
> |  1   | 0.6377 |  47.20  |
> |  2   | 0.6382 |  47.01  |
> |  3   | 0.6377 |  46.62  |
> |  4   | 0.6381 |  46.75  |
> |  5   | 0.6379 |  47.11  |
> | Avg. | 0.6379 |  46.94  |
> | Std. | 0.0002 |  0.219  |
>
> These results demonstrate minimal variability in both loss and accuracy, with a particularly low standard deviation for the loss (0.0002). This confirms that our training process is robust and not significantly affected by randomness. Furthermore, the fluctuations in loss are smaller than those in accuracy on downstream tasks. Therefore, we used the loss as a stable metric to estimate the parameters of the scaling law function.
>
> **(2) Predictive Validation of the Scaling Law**:
>
> To further validate the reliability of our scaling law, we performed predictive validation by holding out the last data point, refitting the scaling function using the remaining points, and then predicting the loss for the hold-out point. The predicted loss was subsequently compared with the actual loss. For example, when fitting the scaling function for the visual embeddings in Section 3.2.1, we excluded the data point at $M=196$ and refitted the curve to obtain $L_M = 0.46, M_0 = 1.04\times 10^{-5}, \alpha_M = 0.09$, with $R^2 = 0.906$. We then use this refitted curve to predict the loss at M=196, obtaining a loss of 0.678, which closesly matches the actual loss of 0.679. We repeat this process for both embedding scaling and frame scaling, and the results are summarized below:
>
> | Setup               | Hold-out point | Predicted loss | Actual loss |
> |:--------------------|:--------------:|:--------------:|:-----------:|
> | Sampling tokens     |   \(M=196\)   |      0.678     |    0.679    |
> | Compressing tokens  |   \(M=196\)   |      0.630     |    0.638    |
> | Sampling frames     |   \(T=128\)   |      0.628     |    0.638    |
> | Compressing frames  |   \(T=128\)   |      0.629     |    0.622    |
>
> These results demonstrate that the predicted losses consistently align with the real losses, with prediction errors below 0.01. This validates the reliability of our scaling law and ensures the robustness of our findings.
>
> In summary, while we acknowledge the constraints of not training multiple models for every data point, the combination of multi-seed robustness tests and predictive validation of our scaling law provides strong evidence of the reliability of our results. These efforts effectively mitigate randomness bias and support the conclusions presented in our study.

---

> ### Author Response · Authors · 2024-11-25
> **Kindly Reminder for the Discussion**
>
> Dear Reviewer 4WaN,
>
> Thanks for your careful reading of our paper. We have made several improvements to our paper based on your feedback. Since the rebuttal stage is coming to an end, we are more than happy to hear your comments and address any of your further concerns during the remaining time.
>
> Best,
>
> Authors

---

> > ### Comment · Reviewer_4WaN · 2024-11-27
> > **thanks for your reply**
> >
> > Thank you for providing additional details and supplementary experiments to further validate the generality of the proposed study. I suggest that the authors incorporate these responses and findings into the final version, as I believe they will significantly strengthen the work.
> >
> > I have no further concerns and have decided to **raise my rating to 8**.

---

> > > ### Author Response · Authors · 2024-11-27
> > > **Appreciate your updated score**
> > >
> > > Dear Reviewer 4WaN,
> > >
> > > We deeply appreciate your time and effort in reviewing our paper and raising your score. Your constructive feedback has been instrumental in enhancing our work, and we are truly grateful for your insightful comments and suggestions.
> > >
> > > Following your advice, we have incorporated the additional details and supplementary findings into the revised version and will ensure they are included in the final version.
> > >
> > > Thank you once again for your time, effort, and support.
> > >
> > > Best,
> > >
> > > Authors

---

### Official Review · Reviewer_UYUt · 2024-11-06

**Soundness:** 2
**Presentation:** 3
**Contribution:** 3
**Rating:** 6
**Confidence:** 4

**Summary:**

This paper mainly discusses the video frame and visual token per frame selection strategies for video MLLMs. Based on LLaVA-like architecture, this paper conducts several experiments, e.g., varying frame number while fixing visual token per frame and altering visual token per frame while fixing total frame number, to fit the loss curves. Finally, it proposes a fitting function of the two factors and draw an optimal combination conclusion.

**Strengths:**

1. The motivation and idea of this paper is straightforward.
2. The writing and structure organization of this paper is clear and easy to follow.
3. The experimental results are clearly clarified, and the statistics are persuasive.

**Weaknesses:**

1. My main concern about this paper is the generalization of the final conclusion. As we all know, video MLLM is very large and complex. This paper uses only one model structure (SigLip + Qwen2-7B) to conduct experiments on several benchmarks. Is the final best setting ($T_{opt} \approx 118$ and $M_{opt} \approx 51$) generalizable to other models and benchmarks? If not, must we conduct several experiments to fit a new curve to find the best factor combination?
2. Based on the above question, if this paper is just an experiment report, readers cannot obtain enough inspiration from this paper.

**Questions:**

Besides the question in Weakness, how can you determine the function formation of Eqs. 3, 4, 5, and 6? Just observe from figures 2a, 3a, and 4?

---

> ### Author Response · Authors · 2024-11-22
> **Official Response to Reviewer UYUt [1/2]**
>
> Thank you for your insightful suggestions. We have outlined our responses to your concerns below. If you have any additional questions, please feel free to let us know. We will do our best to address them.
>
> > **W1**: My main concern about this paper is the generalization of the final conclusion. As we all know, video MLLM is very large and complex. This paper uses only one model structure (SigLip + Qwen2-7B) to conduct experiments on several benchmarks. Is the final best setting (Topt≈118 and Mopt≈51) generalizable to other models and benchmarks? If not, must we conduct several experiments to fit a new curve to find the best factor combination?
>
> **R1**: Thanks for the insightful question. Although the video MLLMs are large and complex, their architectures have been gradually converging toward a common structure: vision encoder+connector+LLM, where the primary components are vision encoder and LLM. To verify the generalization of our conclusion in Section 4, we utilize other representative base models: **LLaMA3-8B** and **CLIP-ViT-L-336px**. Experiments are conducted under the same settings as described in Section 4. The experimental results in the following table show that both the minimum loss and the optimal benchmark performance are achieved with 120 frames and 49 visual embeddings per frame, consistent with the conclusion in Section 4. This demonstrates the generalizability of our conclusion and is expected to inspire future developments in video MLLMs.
>
> | #Frames | # Embed./ Frames | # Total Embed. | Loss↓ | Event-Bench | VNBench | MLVU  | LongVideoBench | VideoMME                          wo/w-subs | Avg.   |
> |:----------:|:-----------------:|:----------------:|:--------:|:-------------:|:---------:|:-------:|:----------------:|:--------------------:|:--------:|
> | 10       | 576             | 5760           | 0.90   | 20.00       | 18.52   | 49.95 | 45.80          | 44.22/43.37        | 36.98  |
> | 30       | 196             | 5880           | 0.91   | *23.33*     | 26.15   | 52.98 | 45.80          | 45.74/46.41        | 40.07  |
> | 72       | 81              | 5832           | 0.89   | 19.33       | 26.67   | 54.99 | 46.03          | 46.59/*47.26*      | 40.15  |
> | 120      | 49              | 5880           | **0.85** | **24.33**   | **30.67** | **57.68** | *47.16*        | *46.93*/47.89      | **42.44** |
> | 162      | 36              | 5832           | *0.86* | 22.00       | *28.74* | *56.80* | **47.99**       | **49.15/49.67**    | *42.39* |
>
> Additionally, Eq. 7 presents a more general conclusion derived from our experiments, and it is easy to generalize across different settings. Specifically, by varying the visual context window length $L$ from 4K to 126K (a range representative of most LLMs), we can compute the corresponding $M_{\text{opt}}$, $T_{\text{opt}}$, and their ratio $\frac{T_{\text{opt}}}{M_{\text{opt}}}$ in the following table.
> | Visual Context Length | # Visual Embed. (M) | # Frames (T) | T / M |
> |:----------------------:|:-------------------:|:------------:|:-----:|
> |        4,000          |         41          |      97      |  2.4  |
> |        6,000          |         51          |     118      |  2.3  |
> |       14,000          |         78          |     178      |  2.3  |
> |       30,000          |        116          |     258      |  2.2  |
> |       62,000          |        169          |     367      |  2.2  |
> |      126,000          |        243          |     517      |  2.1  |
>
> To verify this conclusion, we conduct additional experiments using Qwen2-7B and SigLIP with a visual context window length of **4K**. The results in the following table show that the lowest loss and optimal performance are achieved with 96 frames and 36 visual embeddings per frame, which closely aligns with the predicted $T=97, M=41$. Due to memory constraints, we will leave the validation of longer context lengths for future work.
>
> | #Frames | # Embed./ Frames | # Total Embed. | Loss ↓  | Event-Bench | VNBench | MLVU  | LongVideoBench | VideoMME wo/w-subs | Avg.   |
> |:--------:|:---------------:|:--------------:|:-------:|:-----------:|:-------:|:-----:|:--------------:|:------------------:|:------:|
> |    18    |       196       |      3528      |  0.8010 |   *25.00*   |  24.81  | 54.75 |      48.68     |     50.15/50.41    |  42.30 |
> |    70    |        49       |      3430      |  0.7581 |     23.00   |  32.52  | 59.60 |   *49.36*      | **54.44/55.33**    | *45.71*|
> |    96    |        36       |      3456      | **0.7539**| **28.00** | *35.63* |**61.64**| **49.51**   | *53.26/53.78*      | **46.97** |
> |   138    |        25       |      3450      | *0.7554* |     22.33   | **37.93**| *60.06*|     47.84     |    52.26/53.19     | *45.60* |

---

> ### Author Response · Authors · 2024-11-22
> **Official Response to Reviewer UYUt [2/2]**
>
> > **W2**: Based on the above question, if this paper is just an experiment report, readers cannot obtain enough inspiration from this paper.
>
> **R2**: Thank you for the question. Actually, prior research on long video understanding has primarily focused on optimizing training data or model architecture, our work highlights a critical yet under-explored problem: ***how to effectively represent a video under a constrained context window?*** We find that the trade-off between the number of frames and visual embeddings has a large impact on the model performance. However, current video MLLMs often neglect it and rely on large-scale hyper-parameters searches for the two factors, which is costly and does not effectively guide the design of video-MLLMs.
>
> Our study aims to fill this gap, investigate the scaling effect of visual context length, and explore the trade-off between the number of frames and visual embeddings. Inspired by prior work on scaling laws [1, 2, 3, 4], we propose a formulation and conduct extensive experiments. The main contribution and insights in our paper are as follows:
>
> - **Scaling law for visual context length**: We present a scaling law that predicts the performance of video MLLMs based on visual context length.
> - **Impact of frame and embedding representation strategies**: We analyze how different strategies for frame and token representation affect the performance of video MLLMs, offering practical insights for optimization.
> - **A principled framework for balancing frames and embeddings**: We propose a general approach for balancing the number of frames and visual embeddings and generalize it to various visual context window lengths in the updated version.
>
> These contributions go beyond reporting experimental setups and results, providing a broader understanding and practical guidelines for designing video MLLMs.
>
> [1] Training Compute-Optimal Large Language Models
>
> [2] Scaling laws for neural language models
>
> [3] Scaling laws for transfer.
>
> [4] Data Mixing Laws: Optimizing Data Mixtures by Predicting Language Modeling Performance
>
> > **Question**: Besides the question in Weakness, how can you determine the function formation of Eqs. 3, 4, 5, and 6? Just observe from figures 2a, 3a, and 4?
>
> The scaling law's general formulation often adopts a power-law function, typically expressed as $L=kx^{\alpha}+c$, where $k, \alpha$, and $c$ are parameters to fit and $x$ can be model parameters or data volume [1,2,3]. We follow this formulation in defining Eq. 3 and Eq. 4. Initially, we applied a power-law form to Eq. 5, as in Eq. 3 and Eq. 4, but the fitted $R^2$ indicated a poor fit. We then shifted to a linear function, which yielded significantly improved fit. This process of testing different functional forms and selecting the best fit has been a standard practice in scaling law studies [3]. For Eq. 6, we utilized a commonly adopted scaling law function to express relationships between two factors, following established approaches from prior work [2,3]. While other functional forms could potentially be explored, the selected equations yielded sufficiently low fitting errors to justify their use.
>
> [1] Training Compute-Optimal Large Language Models
>
> [2] Scaling laws for neural language models
>
> [3] MiniCPM: Unveiling the Potential of Small Language Models with Scalable Training Strategies

---

> ### Author Response · Authors · 2024-11-25
> **Kindly Reminder for the Discussion**
>
> Dear Reviewer UYUt,
>
> Thanks for your careful reading of our paper. We have made several improvements to our paper based on your feedback. Since the rebuttal stage is coming to an end, we are more than happy to hear your comments and address any of your further concerns during the remaining time.
>
> Best,
>
> Authors

---

> ### Author Response · Authors · 2024-11-28
>
> Dear Reviewer UYUt,
>
> We sincerely appreciate the time and effort you have dedicated to reviewing our paper. In response to your comments, we have carefully conducted additional experiments and revised our paper accordingly:
>
> 1. We verified the optimal settings using other representative base models (LLaMA3-8B and CLIP-ViT-L-336px).
>
> 2. We extended our analysis to generalize the conclusion across visual context lengths ranging from 4K to 128K.
>
> We would be grateful if you could let us know whether our revisions and responses address your concerns, or if there are additional questions or clarifications you would like us to provide. Since the rebuttal stage is coming to an end, we are more than happy to hear your comments and address any of your further concerns during the remaining time.
>
> Best,
>
> Authors

---

> ### Author Response · Authors · 2024-12-03
> **Kindly Reminder for the Discussion**
>
> Dear Reviewer UYUt,
>
> We sincerely appreciate your time and efforts in reviewing our paper. We have carefully addressed the suggestions and concerns raised by the reviewers. Specifically, we verified our conclusions using other representative base models and extended them to accommodate various visual context lengths.
>
> As the rebuttal deadline approaches, we kindly seek your feedback on the revised version of the paper. Your insights are invaluable in improving our work, and we greatly appreciate your timely response.
>
> Thank you once again for your time and thoughtful review.
>
> Best regards,
>
> Authors

---

### Author Response · Authors · 2024-11-22
**Revision of our Manuscript**

We sincerely thank the reviewers for their thoughtful feedback and constructive suggestions, which greatly help us improve our work. In this work, we aim to explore an important but under-explored problem for video-MLLM, i.e., the design of visual context encoding part, and conduct comprehensive experiments to formulate useful rules to determine its design. We follow the reviewer's suggestion and update the paper, and the main revision contents are listed as follows:

- Figure 2 (b) and Figure 3 (b): We add the averaged performance in them.
- Appendix B: We add the experiment on other LLM and vision encoder backbones.
- Section 4 and Appendix C: We generalize our conclusion to various video context lengths.
- Appendix D: We add the expriments of scaling visual tokens in the vision encoder and compare it with scaling visual tokens input to LLM.
- Appendix E: We add the experiments under different seeds and the predictive validation of the scaling law.
- Appendix F: We scale the video data and compare our model with several SOTA video MLLMs.
- Appendix G: We add the experiments without any frames and visual embeddings as a reference.

We hope that the above works are able to address the reviewers' concerns, and are also very willing to have more discussion with all reviewers.

---

### Author Response · Authors · 2024-11-27

Dear Reviewers,

Thank you for your time and effort in reviewing our manuscript. We sincerely appreciate the thoughtful feedback you provided, which has been invaluable in improving our work.

In our rebuttal, we have made a concerted effort to address your concerns through detailed responses, supplementary experiments, and revisions to the manuscript. As the discussion phase draws to a close, we would be very grateful if you could take a moment to review our response.

If you find that our revisions have adequately addressed your concerns, we kindly ask if you might consider reevaluating your initial assessment and adjusting the score accordingly. Should there be any remaining issues or points requiring further clarification, we are more than happy to address them promptly.

Thank you again for your valuable time and insights.

Best regards,

Authors

---

### Comment · Area_Chair_L4CZ · 2024-11-27
**Please take a look at the authors' rebuttal and start a discusssion if needed**

Dear Reviewers,

Thanks for your contributions in reviewing this paper.

As the author-reviewer discussion deadline is approaching, please could you take a look at the authors' rebuttal (if not yet) and see if it addressed your concerns or if you have any further questions. Please feel free to start a discussion.

Thanks,

AC

---

### Meta-Review · Area_Chair_L4CZ · 2024-12-18

**Metareview:**

In this paper, the authors presented a study about the design space of visual context representation in video multimodal large language models (MLLMs). Specifically, two components were considered for the analysis: the number of frames and the number of embeddings/tokens per frame. By exploring the scaling effects within these aspects, empirical findings were presented with an optimal formula for determining the two components. The main strengths of this paper are:
- The paper is well motivated and the idea is interesting, offering a valuable perspective for visual context representation design in video MLLMs. The idea is simple and straightforward, with a clear presentation.
-  Extensive experimental analysis was presented, showing several interesting and useful findings, e.g. the trade-off between embeddings and frames, the contribution of mean pooling vs. uniform sampling, etc. The analysis presented in this paper could be useful for follow-up research in the community.
- The paper is well-written, well-organised, and easy to follow.

The main weaknesses of the paper were mainly around the generalisation ability, the impact of the corresponding derived conclusions, unclear analysis of the contributions from different components, and comparison with state-of-the-art.

After the authors' rebuttal, followed by a few rounds of discussions, the above major concerns were addressed, and the reviewers were satisfied with the supplemented evidence. Two reviewers decided to raise their ratings, and the paper ended up with 1 Accept and 2 borderline Accept. Considering the consistent positive ratings and the potential contributions this paper could bring to the community, the AC is happy to recommend an Accept.
But the authors are strongly recommended to incorporate the additional experiments and discussions into their final version.

**Additional Comments On Reviewer Discussion:**

During the rebuttal and discussion period, the authors provided responses and additional evidence to answer the raised questions and concerns from the reviewers. The reviewers engaged in the discussion phase and replied with their thoughts about the rebuttal and comments from other reviewers. For example, the rebuttal addressed the concern about the generalisation ability from reviewer 4WaN, who finally raised their rating; the further experiments and the "blind" MLLM addressed the concerns from reviewer WRue. Reviewer L4CZ was also satisfied with the rebuttal and raised the rating.

In general, two out of three reviewers raised their rating and the paper overall received a positive rating. But at the same time, the authors were strongly recommended to include the further results and discussions in their final version, to clear those concerns and strengthen the paper.

---

### Decision · Program_Chairs · 2025-01-22

Accept (Poster)